# Evaluation of Reference Genes Suitable for Gene Expression during Root Enlargement in Cherry Radish Based on Transcriptomic Data

Yao Yao [1], Xiaoqian Wang [1], Bingxing Chen [1], Shurui Zheng [1], Gefu Wang-Pruski [1,2], Xiaodong Chen [1,3,*] and Rongfang Guo [1,3,*]

[1] Joint FAFU-Dalhousie Lab, College of Horticulture, Fujian Agriculture and Forestry University, Fuzhou 350002, China

[2] Department of Plant, Food, and Environmental Sciences, Faculty of Agriculture, Dalhousie University, Truro, NS B2N 5E3, Canada

[3] Institute of Horticultural Biotechnology, College of Horticulture, Fujian Agriculture and Forestry University, Fuzhou 350002, China

[*] Correspondence: xdchen007@163.com (X.C.); guorofa@163.com (R.G.)

**Abstract:** Reliable reference genes (RGs) are of great significance for the normalization of quantitative data. RGs are often used as a reference to ensure the accuracy of experimental results to detect gene expression levels by reverse transcription–quantitative real-time PCR (RT-qPCR). To evaluate the normalized RGs that are suitable for studying the expression of genes during the process of radish stele enlargement, based on the functional annotations and fragment per kilobase of transcript per million mapped reads (FPKM) values in the transcriptome data, three traditional RGs (*GAPDH*, *18SrRNA*, and *ACTIN7*) and seven commonly used RGs (*UBQ11*, *TUA6*, *TUB6*, *EF-1b1*, *EF-1a2*, *PP2A11*, and *SAND*) were obtained. In the study, the results of geNorm, NormFinder, and BestKeeper from RefFinder comprehensively analyzed the stability ranking of candidate RGs. The results showed that compared with the traditional RGs, the common RGs show higher and more stable expression. Among the seven commonly used RGs, *PP2A11* is recommended as the optimal RG for studying cherry radish stele enlargement. This research provides a useful and reliable RG resource for the accurate study of gene expression during root enlargement in cherry radishes and facilitates the functional genomics research on root enlargement.

**Keywords:** cherry radish; root enlargement; stele; reference genes; *PP2A11*

## 1. Introduction

Cherry radish (*Raphanus sativus* L. var. *radculus* pers) belongs to the herbaceous plant and is a new type of root vegetable. The fleshy root is the storage and edible organ of the cherry radish, and the development of the fleshy root is closely related to the yield and quality of the cherry radish [1]. Thus, it is of great significance to study the development mechanism of fleshy roots in cherry radishes. Plant physiology studies have shown that the stele is the main part of the radish's expanding fleshy root.

The formation of fleshy roots is the result of cortex rupture and stele expansion [2]. Little research has been reported on the genes involved in the regulation of radish fleshy roots. Yu et al. (2016) used transcriptome to reveal a complex regulatory network in the thickening of radish taproots [3]. The study concluded that genes including *AUX/IAA* [4], *ARF* [5], and the corresponding miRNAs play an important role in the regulation of fleshy root expansion in radishes. The detection of gene expression levels is essential to unravel the mechanism of root expansion. The selection of appropriate RGs to normalize the expression levels of the genes tested is crucial for the accuracy and reliability of qRT-PCR data [6]. To assess the relative expression of genes, it is common to select applicable

normalized RGs. RGs are genes that are stably expressed at different stages of development, in different parts of the plant, and under different physiological conditions [7]. Generally, the housekeeping genes including *TUA* (*Tubulin alpha*), *TUB* (*Tubulin beta*), *EF-1a* (*Elongation factor 1-alpha*), *ACTIN*, *GADPH* (*Glyceraldehyde-3-phosphate dehydrogenase*), *18SrRNA*, and *UBQ* (*Ubiquitin*) are selected as the RG; housekeeping genes refer to genes that maintain the basic metabolism of cells, and their expression is relatively stable in all tissues and cells [8]. Suitable RGs have been identified in model plants such as *Arabidopsis thaliana* [9] and rice (*Oryza sativa* L.) [10]. However, studies have shown that the expression of RGs is not always constant across tissues and is varied in different conditions, or even varies greatly, and no RGs have yet been found to be consistently expressed under all conditions [11]. Kim et al. found that *18SrRNA* was very stable in gene expression studies at different developmental stages in rice [12], while Nitin et al. (2018) do not recommend the *18SrRNA* gene as an RG for daytime/circadian rhythm samples [13]. This showed there are significant differences in the expression of the RGs under different conditions.

To further elucidate the regulatory mechanism of fleshy root enlargement in cherry radishes, it is important to identify the normalized RGs for gene expression during root enlargement. Researchers have identified some RGs in radish. Using radish-advanced inbred lines, Xu et al. analyzed the relatively stable expression of genes in different tissue parts, especially in leaves under different conditions, and concluded that RPII (RNA polymerase-II transcription factor), *TEF2* (translation elongation factor 2), and *ACTIN* are suitable as internal reference genes for the quantitative analysis of radish genes [14]. Duan et al. collected the flower buds and siliques of radishes at different reproductive stages and obtained *UP2* (uncharacterized conserved protein UCP022280) and *GAPDH* (glyceraldehyde-3-phosphate dehydrogenase) as suitable RGs for radish pistil development studies [15]. The selection of RG was also conducted in different organs including root, stem, and leaf, as well as calyx, petal, stamen, and pistil and UPR (uncharacterized protein family), *GSNOR1* (GroES-like zinc-binding dehydrogenase family protein), and *ACTIN2/7*, which were the most stable internal reference genes in radish [15]. However, no RGs are selected for studying the enlargement of radish root. The fleshy root of the radish mainly develops from the stele [16].

In the current study, the enlarging part of the cherry radish (stele) was used to screen the RGs for analysis of the gene expression during the enlargement of the radish. We have previously observed the expansion of the cherry radish root by microscopy of the cross-section and obtained the transcriptome data of key points of radish enlargement. Based on the transcriptomic data, 10 RGs with different functions (*GAPDH*, *ACTIN*, *18SrRNA*, *UBQ*, *TUA*, *TUB*, *EF-1a*, *EF-1b*, *PP2A11*, and *SAND*) were screened as candidate RGs. A comprehensive analysis of the candidate RGs' stability was conducted using RT-qPCR technology and NormFinder, geNorm, and BestKeeper from RefFinder [17]. The suitable RGs were evaluated to analyze the expression of related genes during cherry radish fleshy root expansion. The identification of reference genes provides a basis for analyzing the expression pattern of genes during radish expansion and facilitates further elucidation of the mechanism of organ expansion in radish products.

## 2. Materials and Methods

### 2.1. Plant Material

The plant cherry radish cultivar is 'Kunyou No. 2', purchased from Beijing Jielia Seed Industry Co., Ltd. (Beijing, China) The cherry radish seeds were sown on a glass petri dish with moist perlite and placed in a constant temperature incubator at a temperature of 28 °C for seed germination. On day 3, the seedlings were moved to a climate chamber for cultivation. On the 9th day, the radish seedlings were transplanted and replaced with nutrient solution. The volume of the nutrient solution was 3 L and the pH value was 6.0. The hydroponic nutrient solution formula included 945 mg·L$^{-1}$ Ca$_2$NO$_3$·4H$_2$O, 809 mg·L$^{-1}$ K$_2$NO$_3$, 153 mg·L$^{-1}$ KH$_2$PO$_4$, 493 mg·L$^{-1}$ MgSO$_4$·7H$_2$O, 30 mg·L$^{-1}$ EDTA-FeNa$_2$, 2.86 mg·L$^{-1}$ H$_3$BO$_3$, 2.13 mg·L$^{-1}$ MnSO$_4$, 0.22 mg·L$^{-1}$ ZnSO$_4$, 0.08 mg·L$^{-1}$ CuSO$_4$, and 0.02 mg·L$^{-1}$ (NH$_4$)$_2$MoO$_4$ [18]. The illumination

conditions were white light for 16 h/dark for 8 h at the 150 μmol·m$^{-2}$·s$^{-1}$ intensity. The temperature was set at 25 °C.

The material for this experiment was the gradually expanding stele of cherry radish. Twelve radish seedlings were planted in one hydroponic container, 30 containers were planted at a time, and from day 9 onwards, three seedlings were taken from each of the three containers for sectioning, and the rest were used for sampling and analysis. The specific observation and sampling processes were as follows: after the radish was transplanted (9 d) until the cherry radish cortex rupture (20 d), the root of the cherry radish was observed daily, and samples were taken, sliced, and microscopically examined to calculate the ratio of the diameter of the radish stele cross-section to the diameter of the root, and thus analyze the expansion of the cherry radish root. In the current study, the samples of stele we selected were 9 d, 12 d, 16 d, and 20 d, and the samples of lateral roots are the same day as the stele. Finally, steles of the 12-day- and 20-day-old cherry radish roots with a ratio of 30% and 70%, respectively, were collected for the qRT-PCR analysis. When sampling the stele, the enlarged part of the radish was broken in the middle and the cortex was split, while the stele remained intact. Then, the external cortex was removed and the stele was cut off with a clean blade, weighed, loaded into a lyophilization tube, and quickly placed in liquid nitrogen. Three biological replicates were performed for each experiment.

### 2.2. Extraction of Total RNA and Synthesis of cDNA

The total RNA was extracted using the RNAiso Plus kit from TaKaRa. The concentration and OD (optical density) value of the RNA were determined by a NanoDrop 2000 ultra-micro spectrophotometer. The quality of the RNA was checked with 1% agarose gel. cDNA was synthesized by using the Evo M-MLV Reverse Transcription Kit (RT Kit with gDNA Clean for qPCRII), which came from Eric Biological Co., Ltd. Then, the purified RNA was digested with gDNA clean reagent to remove genomic DNA at 42 °C following the 10 μL reaction system including 1 μL of gDNA clean reagent, 2 μL of 5xgDNA clean buffer, 1 μg of total RNA, and RNase-free water. Next, the cDNA was synthesized according to the manufacturer's system including 1 μL of Evo M-MLVRTase enzyme mix, 1 μL of RT primer mix, 4 μL of 5xRTase reaction buffer Mix I, and 4 μL of RNase-free water, under reaction conditions of 37 °C, 15 min, and 85 °C, 5 s. RNA extraction and cDNA synthesis from all samples was performed with three biological replicates. The experiment was carried out in a clean and dust-free fume hood. The researchers were required to wear rubber gloves and masks during the experiment to avoid contamination of the RNA.

### 2.3. Screening of Candidate RGs and Primer Design

The genomic data of the radish came from the National Center for Biotechnology Information (https://www.ncbi.nlm.nih.gov/genome/12929, accessed on 10 July 2022). Transcriptome sequencing of the candidate RG was obtained using BioProject ID PRJNA874325 (https://dataview.ncbi.nlm.nih.gov/object/PRJNA874325?reviewer=r6vc02uh53bundl6rarngrsds, accessed on 10 July 2022). Based on the previous studies on cruciferous RGs, we performed gene function annotation on the RGs in transcriptome data and initially screened out 23 different functional candidate RGs. By analyzing the gene FPKM values, and through general PCR analysis, we found that the FPKM values of 13 of these RGs were not high (Table 1), and running PCR with mixed samples revealed that the bands were not bright, and finally 10 RGs were identified as candidates. Gene CDS sequences were obtained and the corresponding primers were designed by Primer Premier 5 software and are listed in Table 1.

**Table 1.** Selected candidate reference genes and their respective PCR primers.

| Primer Number | Gene ID | Gene Annotation | FPKM Value | Gene Description | Primer Sequences (5'-3') | Tm (°C) | Length of Product (bp) |
|---|---|---|---|---|---|---|---|
| 1 | RSG46678 | *GAPDH* | 5.37 | Glyceraldehyde-3-phosphate dehydrogenase | F: TTGCCGTCTCCAGAATCCCT<br>R: CGTGCCAACACCTGAGGAAGT | 61.7<br>62.3 | 285 |

**Table 1.** *Cont.*

| Primer Number | Gene ID | Gene Annotation | FPKM Value | Gene Description | Primer Sequences (5′-3′) | Tm (°C) | Length of Product (bp) |
|---|---|---|---|---|---|---|---|
| 2 | RSG01140 | *18SrRNA* | 29.56 | 18S ribosomal RNA | F: GACTCAATCGTCCAGAAAGCAG<br>R: CAAGCGGTTAGAAAGGGGAG | 59.1<br>59.2 | 186 |
| 3 | RSG23795 | *UBQ11* | 134.49 | Ubiquitin 11 | F: CATCTCGTTCTCAGGCTTCG<br>R: CAATGTTCTACCGTCCTCAAGC | 58.2<br>59.1 | 195 |
| 4 | RSG21557 | *TUA6* | 397.2 | Tubulin alpha 6 | F: GGTATCCAGGTCGGAAATGC<br>R: CGTCGATCACAGTGGGCTCT | 59.2<br>60.6 | 193 |
| 5 | RSG05570 | *TUB6* | 374.36 | Tubulin beta 6 | F:ACTTCGTTTTCGGGCAATCT<br>R: CCTTAGGTGATGGGAAGACAGAG | 59<br>59.7 | 258 |
| 6 | RSG42380 | *EF-1b1* | 633.74 | Elongation factor 1-beta | F: ATCACTGTCTTTGCTGCTCTTGC<br>R: TTCTTCTCCTCCTCGGTCTCC | 61.4<br>59.8 | 260 |
| 7 | RSG33614 | *EF-1a2* | 524.19 | Elongation factor 1-alpha | F: AAGATGGATGCTACTACCCCTAAG<br>R: CACTGGCACCGTTCCAATAC | 58.5<br>58.6 | 297 |
| 8 | RSG21210 | *PP2A11* | 94.92 | Threonine-protein phosphatase 2A | F: GCTTCCTGGGCTGATTTCGT<br>R: AGCTTTCTTTGTGCCTTGGTC | 61.6<br>58.8 | 159 |
| 9 | RSG39901 | *ACTIN7* | 24.38 | Actin protein | F: CTGAGGACGAACTTGCTTACGA<br>R: CAGTGTTCTCCAAGAGTTGCCTAT | 59.9<br>59.7 | 225 |
| 10 | RSG23793 | *SAND* | 12.51 | Protein SAND | F: TGAAGGTGGATTGCGTGTTG<br>R: CATAGAGTTTCTGGTATGCTCGGTA | 59.9<br>60.1 | 245 |

### 2.4. Specificity Detection of Candidate RGs Primers

Equal amounts of cDNA from all samples were mixed to construct a library. PCR amplification was performed according to the following procedure with pre-denaturation at 94 °C for 3 min; denaturation at 94 °C for 30 s, annealing at 60 °C for 30 s, and extension at 72 °C for 10 s for 35 cycles. The reaction system contained 10 μL of 2xTaq Master Mix 10, 1 μL of forward primer, 1 μL of reverse primer, 7 μL of ddH$_2$O, and 1 μL of cDNA template. PCR products were tested by 1% agarose gel electrophoresis for primer specificity.

### 2.5. Analysis of RT-qPCR and Amplification Efficiency

RT-qPCR was performed using an SYBR® Green Pro Taq HS qPCR Kit (ACCURATE BIOTECHNOLOGY(HUNAN, CHINA)CO.,LTD). The reaction system includes 10 μL of 2xSYBR Green Pro Taq HS Premix* 6, 2 μL of cDNA Template*, 0.4 μL of 10 μM forward and reverse primer, respectively, and 7.2 μL RNase-free water. The reaction condition was followed by pre-denaturation at 95 °C for 30 s, denaturation at 95 °C for 5 s, and annealing at 60 °C for 30 s, with 40 cycles. The dissociation stage reaction condition procedure was set according to Bio-RAD CFX Manager 3.0 (3.0.1224.1015). After the reaction, the gene specificity was analyzed using the obtained melting curve. To obtain the standard melody, the mixed cDNA of all samples was diluted at a ratio of cDNA to RNA-free water at 10:34, and then diluted into cDNA with five concentration gradients at the same dilution ratio. The diluted cDNA was used as a template for the RT-qPCR experiments. The primer amplification efficiency was calculated by E = $(10^{-1/\text{slope}} - 1) \times 100\%$, and 'slope' is the slope of a standard curve.

### 2.6. Data Analysis

The expression stability of the candidate genes was evaluated by BestKeeper [19], NormFinder [20], and geNorm [21] from RefFinder [13,22]. The analysis principle of geNorm is calculating the stability value M by entering the Ct value [23]. The criterion is that the smaller the M value, the better the stability of the reference genes; otherwise, the worse the stability of the reference genes. The software NormFinder was used to analyze the stability of the internal reference genes [20]. The stable value was calculated and the variation between samples was analyzed. BestKeeper software was used to determine the stability ranking of the reference genes by analyzing the correlation coefficient, standard deviation, and coefficient of variation among the genes. The greater the correlation coefficient, the smaller the standard deviation and the coefficient of variation, showing the better the stability of the reference gene, and vice versa.

Data processing and mapping was conducted using WPS and GraphPad Prism software. The significant difference was based on a one-to-one comparison using SPSS software.

## 3. Results

### 3.1. Analysis of RNA Quality, Primer Specificity, and Amplification Efficiency

The concentration of RNA is 445.2 ng·μL$^{-1}$–2277.7 ng·μL$^{-1}$, OD (260 nm/280 nm) value is 1.87–2.12. Electrophoresis detection showed that there are two bright bands, 28S and 18S, respectively (Figure 1A). The above results indicated that the extracted RNA was of good quality, which complied with the rules of quantitative RT-PCR [24].

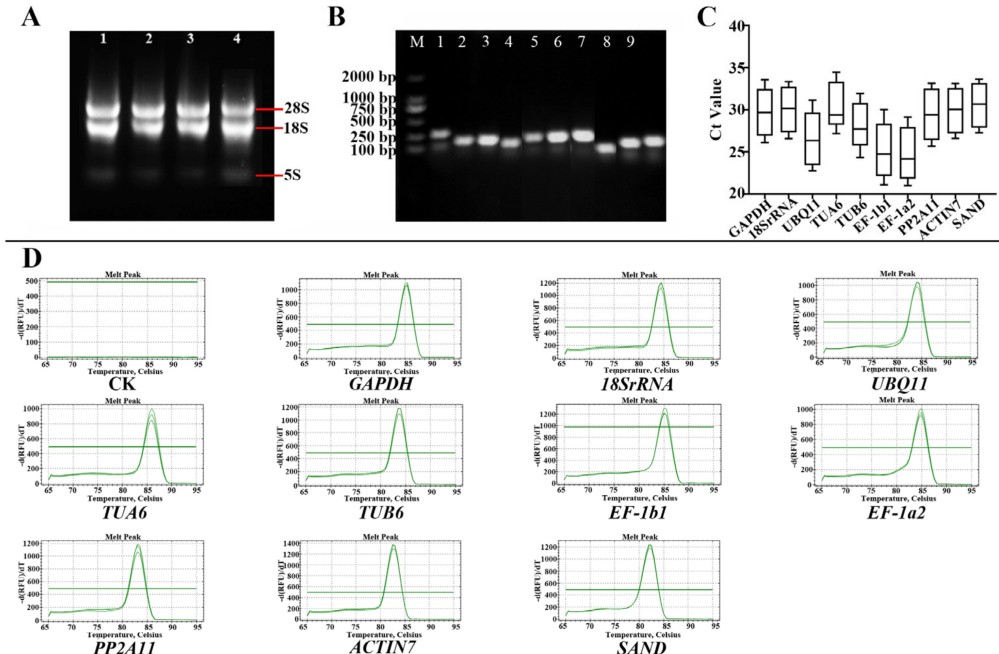

**Figure 1.** Specificity detection of primer amplification and gene expression analysis of candidate reference genes. (**A**) RNA gel figure of stele. 1–4, stele samples; (**B**) primer amplification specificity and amplification length of candidate RGs. M: marker; 1–10: ten candidate RGs in order are, *GAPDH*, *18SrRNA*, *UBQ11*, *TUA6*, *TUB6*, *EF-1b1*, *EF-1a2*, *PP2A11*, *ACTIN7*, and *SAND*; (**C**) boxplot of Ct values of candidate internal reference genes. In the boxplot, each box represents the Ct value of the same reference gene at different concentrations. The horizontal line in the box represents the median, the upper and lower lines represent the upper and lower quartiles, and the upper and lower edges represent the maximum and minimum values of this group of data; (**D**) melting curves of the candidate reference gene. CK means control. Ct means cycle threshold.

The primer-specific electrophoresis results of the candidate RGs showed that the product bands of the 10 candidate RGs were clear and consistent with the length of the designed biological target product (Figure 1B). The Ct values and melting curves were obtained by Bio-Rad CFX Manager 3.0 (3.0.1224.1015) software (Figure 1D). The melt curves of the 10 candidate RGs all had a single peak value and good repeatability, which indicated that the primer specificity was good, and could be used in fluorescence RT-qPCR experiments. Taking the cycle threshold (Ct) value as ordinate and log (C, dilution multiple) values as abscissa, the slope of the standard curve was obtained (Table 2). The results showed that the slope of the standard curve is between −2.14 and −1.69, and the absolute value of the correlation coefficient (r) is in the range of 0.965–0.999. This result showed a high degree of correlation between the Ct value and the log (C, dilution factor) value. The correlation coefficient (r$^2$) is greater than 0.932, close to 1, indicating that the result is highly reliable. The amplification efficiency (E) ranged from 1.001 to 1.398, indicating good amplification efficiency (Table 2).

**Table 2.** Amplification correlation coefficients and amplification efficiency rates of candidate RGs.

| Number | Gene Name | Slope of Standard Curve | Pearson Moment Correlation (r) | $r^2$ | Amplification Efficiency (E) |
|---|---|---|---|---|---|
| 1 | *GAPDH* | −1.86 | −0.999 | 0.998 | 1.220 |
| 2 | *18SrRNA* | −1.76 | −0.996 | 0.992 | 1.321 |
| 3 | *UBQ11* | −2.09 | −0.993 | 0.986 | 1.030 |
| 4 | *TUA6* | −1.74 | −0.965 | 0.932 | 1.340 |
| 5 | *TUB6* | −1.77 | −0.984 | 0.968 | 1.306 |
| 6 | *EF-1b1* | −2.14 | −0.990 | 0.980 | 1.001 |
| 7 | *EF-1a2* | −2.04 | −0.994 | 0.988 | 1.065 |
| 8 | *PP2A11* | −1.97 | −0.995 | 0.990 | 1.123 |
| 9 | *ACTIN7* | −1.73 | −0.993 | 0.986 | 1.354 |
| 10 | *SAND* | −1.69 | −0.991 | 0.982 | 1.398 |

*3.2. Ct Value Analysis of Candidate RGs*

The Ct value refers to the number of cycles for the fluorescent signal to reach the set fluorescence threshold during the detection process of qRT-PCR technology. The smaller the Ct value, the higher expression abundance. To analyze the distribution differences in the expression of each candidate RG, the expression levels of ten candidate RGs were analyzed by qRT-PCR. Boxplots were drawn with GraphPad Prism software (Figure 1C). The results showed that the differences in the distribution range of the Ct values of the 10 RGs were between 6.34 and 8.92, among which *SAND* showed the smallest change and *EF-1b1* showed the largest change. The Ct values of the candidate RGs are between 20.98 and 34.47, among which the Ct value of *EF-1a2* is the smallest, that is, the highest expression level, and the Ct value of *TUA6* is the largest, that is, the lowest expression level. The results are consistent with the results in Figure 1B. To compare the expression characteristics of ten candidate RGs under different experimental conditions and obtain the best RG under a specific condition, we evaluated the stability of ten candidate RGs in combination with specific experimental conditions.

*3.3. Stability Analysis by geNorm, NormFinder, and BestKeeper*

The website RefFinder (http://blooge.cn/RefFinder/?type=reference, accessed on 12 August 2022), which contains calculation results of the geNorm, NormFinder, and BestKeeper, has been widely used in the evaluation of RGs to analyze the stability of the RGs. The edible part of the radish is the fleshy taproot, which is formed mainly by the enlargement of a stele. In this study, two groups of experimental materials were set up: the first group was the stele (Group I) and the second group was the lateral root (Group II), to analyze the differences of normalized stable RGs in the fleshy taproot (stele) and lateral root development.

geNorm is software that calculates the stability M value of each RG by entering the Ct value into the program to filter out the RGs with better stability [25]. geNorm (M) provides a comparison of stability as the average pairwise variation between the gene and all other candidate genes [23]. Therefore, the principle of its judgment is that if the gene exhibits a high degree of variability among samples, i.e., the higher the M value, the less stable the gene is, the less suitable it is for RG selection, and vice versa, the better it is for RG selection. The results in geNorm showed that the stability values (M) of *PP2A11* are the smallest in both two groups and the gene shows a low degree of inter-sample variability, so it was ranked first in terms of stability (Figure 2A), which is suitable for RG selection.

NormFinder focuses on finding genes that have less variation in expression within and between groups, and then the gene with the least variation is measured. The criteria for the NormFinder program are similar to those for the geNorm program. The principle is that the smaller the stability M-value, the more stable the gene expression is, and vice versa [26]. In the paper, the analysis of NormFinder also showed that the lowest stability values were *PP2A11* and *18SrRNA* in both groups (Figure 2B).

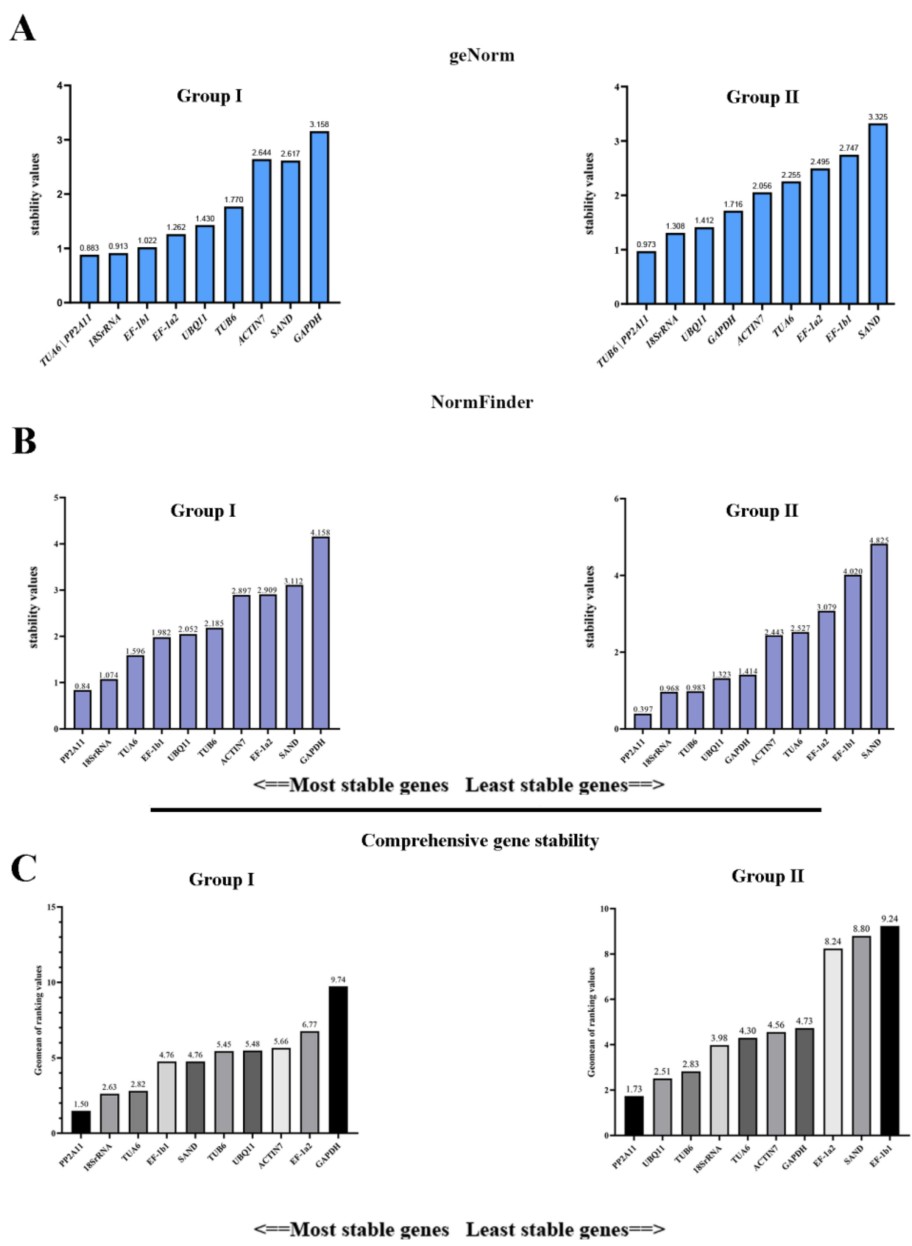

**Figure 2.** Stability analysis of candidate RGs. (**A**) Stability analysis in geNorm; (**B**) expression of stable value in NormFinder; (**C**) comprehensive ranking of gene stability.

BestKeeper is a method based on stability index (BKI) calculations that indicate the highest stability [19]. This software also performs the analysis by directly entering the Ct values of the samples. By calculation, BestKeeper can obtain the correlation coefficient (r), standard deviation (SD), and coefficient of variation (CV) between each gene to generate pairs and then compare the values to calculate the stability value [27]. The criterion principle is that the larger the correlation coefficient (r), the smaller the standard deviation (SD) and the coefficient of variation (CV), and the better the stability value of the internal reference genes, and vice versa [19]. In this paper, according to the correlation coefficient (R), standard deviation (SD), and coefficient of variation (CV), the stability values were obtained by BestKeeper (Table 3). The results showed that *SAND* was the most stable gene in Group I, *EF-1a2* was the least stable gene, and *PP2A11* ranked the fifth. *TUA6* was the most stable gene in Group II, *EF-1b1* was the least, and *PP2A11* ranked the third.

**Table 3.** Stability analysis of BestKeeper.

| Gene Name | Group I | | | | | Group II | | | | |
|---|---|---|---|---|---|---|---|---|---|---|
| | Correlation Coefficient (r) | Standard Deviation (SD) | Coefficient of Variation (CV) | Gene Stability | Ranking | Correlation Coefficient (r) | Standard Deviation (SD) | Coefficient of Variation (CV) | Gene Stability | Ranking |
| *GAPDH* | 0.455 | 3.84 | 11.93 | 3.837 | 9 | 0.925 | 3.41 | 11.64 | 3.408 | 4 |
| *18SrRNA* | 0.982 | 3.05 | 9.34 | 3.053 | 4 | 0.968 | 3.57 | 11.89 | 3.57 | 7 |
| *UBQ11* | 0.946 | 3.22 | 10.99 | 3.222 | 6 | 0.969 | 3.44 | 13.04 | 3.442 | 5 |
| *TUA6* | 0.992 | 3.62 | 10.75 | 3.615 | 7 | 0.815 | 1.99 | 6.46 | 1.994 | 1 |
| *TUB6* | 0.832 | 2.93 | 9.20 | 2.934 | 3 | 0.995 | 3.74 | 12.94 | 3.74 | 8 |
| *EF-1b1* | 0.963 | 3.69 | 12.48 | 3.69 | 8 | 0.935 | 5.73 | 20.2 | 5.727 | 10 |
| *EF-1a2* | 0.990 | 4.42 | 14.71 | 4.422 | 10 | 0.984 | 5.4 | 19.86 | 5.397 | 9 |
| *PP2A11* | 0.991 | 3.14 | 9.52 | 3.139 | 5 | 0.980 | 5.23 | 10.78 | 3.232 | 3 |
| *ACTIN7* | 0.792 | 0.20 | 0.73 | 0.199 | 2 | 0.828 | 2.71 | 8.12 | 2.712 | 2 |
| *SAND* | 0.053 | 0.18 | 0.67 | 0.179 | 1 | 0.257 | 3.47 | 11.01 | 3.468 | 6 |

### 3.4. Comprehensive Analysis of Candidate RGs

The results of the comprehensive analysis of RG stability showed that the stability of RG expression was different in different experiments (Figure 2C). The top five genes for overall stability in Group I were *PP2A11* > *18SrRNA* > *TUA6*> *EF-1b1* > *SAND,* and the top five genes for overall stability in Group II were *PP2A11* > *UBQ11* > *TUB6* > *18SrRNA* > *TUA6.* The RGs with high stability in the two groups were *PP2A11* and *18SrRNA*, with *PP2A11* ranking first (Table 4). The above results showed that *PP2A11* matches the characteristics of high expression level and good stability as an RG. Moreover, traditional gene *18SrRNA* also showed good comprehensive stability in Group I. In addition, the results showed that the other two traditional RGs (*GAPDH* and *ACTIN7*) showed poor comprehensive stability. It can be seen that the traditionally used RGs are not always suitable for all experimental conditions, which also reflects the necessity to screen suitable RGs under different conditions.

**Table 4.** Comprehensive ranking of internal RGs stability.

| Sample Groups | Gene Name | Ge Norm | NormFinder | BestKeeper | Delta CT | Geomean of Ranking Values | Recommended Comprehensive Ranking |
|---|---|---|---|---|---|---|---|
| Group I | *PP2A11* | 1 | 1 | 5 | 1 | 1.50 | 1 |
| | *18SrRNA* | 3 | 2 | 4 | 2 | 2.63 | 2 |
| | *TUA6* | 1 | 3 | 7 | 3 | 2.82 | 3 |
| | *EF-1b1* | 4 | 4 | 8 | 4 | 4.76 | 4 |
| | *SAND* | 9 | 9 | 1 | 9 | 5.20 | 5 |
| | *TUB6* | 7 | 6 | 3 | 7 | 5.45 | 6 |
| | *UBQ11* | 6 | 5 | 6 | 5 | 5.48 | 7 |
| | *ACTIN7* | 8 | 8 | 2 | 8 | 5.66 | 8 |
| | *EF-1a2* | 5 | 7 | 10 | 6 | 6.77 | 9 |
| | *GAPDH* | 10 | 10 | 9 | 10 | 9.74 | 10 |
| Group II | *PP2A11* | 1 | 1 | 3 | 3 | 1.73 | 1 |
| | *UBQ11* | 4 | 2 | 5 | 1 | 2.51 | 2 |
| | *TUB6* | 1 | 4 | 8 | 2 | 2.83 | 3 |
| | *18SrRNA* | 3 | 3 | 7 | 4 | 3.98 | 4 |
| | *TUA6* | 7 | 7 | 1 | 7 | 4.30 | 5 |
| | *ACTIN7* | 6 | 6 | 2 | 6 | 4.56 | 6 |
| | *GAPDH* | 5 | 5 | 4 | 5 | 4.73 | 7 |
| | *EF-1a2* | 8 | 8 | 9 | 8 | 8.24 | 8 |
| | *SAND* | 10 | 10 | 6 | 10 | 8.80 | 9 |
| | *EF-1b1* | 9 | 9 | 10 | 9 | 9.24 | 10 |

The determination of the number of RG combinations is conducive to obtaining more accurate and reliable data. The pairwise variation value (V) of the RGs was calculated by geNorm software. Taking the relationship between the value of Vn/Vn + 1 and the value of 0.15 as the criterion, when Vn/Vn + 1 < 0.15, the number of internal RG combinations is n. When Vn/Vn + 1 > 0.15, the number of internal RG combinations is n + 1, and just evaluate the V2/3 value. The calculation results showed that V2/3 (Group I) = 0.130 < 0.15 (Figure 3A), V2/3 (Group II) = 0.443 > 0.15 (Figure 3B). Thus, the number of RG combinations recommended are two and three, respectively.

A

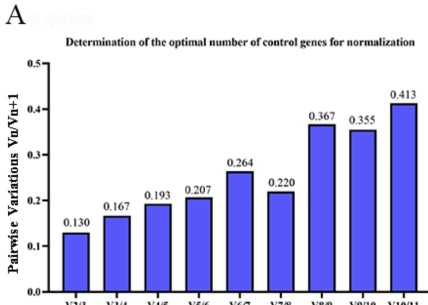

B

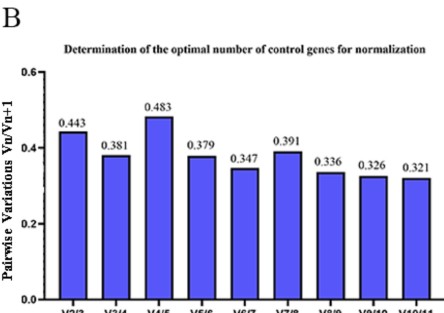

**Figure 3.** Pairing number variation of RGs. (**A**) Pairwise variations of RGs of Group I; (**B**) pairwise variations of RGs of Group II.

### 3.5. Validation of RGs Expression Canonicalization of B-Type RRs

In dicotyledonous plants, the roots and stems of plants are thickened by cell proliferation in the cambium [28]. Cytokinin is an important phytohormone that regulates the activity of the cambium. In the primary root, either the disruption of the expression of the cytokinin receptor or the depletion of cytokinin content by overexpressing cytokinin oxidase/dehydrogenase (CKX) genes prohibits periclinal cell division of pro-cambium cells and reduces the size of the vasculature [29]. Type-B *Arabidopsis* response regulators (*ARRs*) mediate primary cytokinin responses and promote cytokinin-induced gene expression [30]. Studies have shown that type-B *ARRs* are transcriptional factors that play a positive role in regulating the gene expression regulated by cytokinins [31]. In *Arabidopsis*, the *arr1 arr10 arr12* triple mutant of type-B *ARRs* shows retarded growth and abnormal vascular bundle development [32]. It is proved that the secondary growth process of radish involves cambium proliferation and the differentiation of secondary conduction tissue [33]. The process of radish expansion is shown in Figure 4A. With the extension of time, the radish's stele area expands laterally at a faster rate, eventually causing the rupture of the cortex on the 20th day and the typical "belly-breaking" phenomenon. Based on transcriptome data, we analyzed the expression trend of *RsRR11-1* (RSG20107), *RsRR11-2* (RSG25754), and *RsRR12* (RSG31240) in cherry radish before and after stele enlargement. The results showed that the expression levels of the three genes were highest when the ratio of the stele to root reached 30%, while the expression levels of the three genes were lowest when the ratio of the stele to root reached 70%, and then the radish exhibited "broken belly" bulging (Figure 4A). Therefore, it is speculated that type-B *RRs* may play an important role in regulating cambium proliferation and vascular bundle differentiation in radishes at the early stage of radish expansion. We verified the reliability of *RsRR11-1*, *RsRR11-2*, and *RsRR12* (primers as shown in Table 5) expression profiles by the recommended RGs. The results showed that when *PP2A11* was used as the normalized RG, the expression of *RsRR11-1*, *RsRR11-2*, and *RsRR12* decreased (Figure 4B). The results confirmed the reliability of *PP2A11* as a normalized RG in the analysis of cherry radish stele enlargement. In addition, *TUA6* and other genes also showed similar expression trends when normalized. However, when normalized by *UBQ11*, the expression trend of *RsRR11-1*, *RsRR11-2*, and *RsRR12* increased significantly in comparison. The expression trend of *RsRR11-2* and *RsRR12*, showed a significant difference from the original expression trend. It showed that *UBQ11* had higher expression variation. The results indicated that there was still variable expression even for relatively stable HGs [34]. To highlight the contrast between *PP2A11* and the rest, Figure 4C contains the same data as Figure 4B with *UBQ11* removed. The results in Figure 4C showed that the differences between *RsRR11-1* and *RsRR11-2* at different periods of the stele development could be shown when nine candidate genes were used as an internal reference, while for *RsRR12*, the expression was significant when three traditional genes (*18SrRNA*, *ACTIN7*, and *GAPDH*) were used as an internal reference. The expression was insignificant when *PP2A11* was calculated as an internal reference.

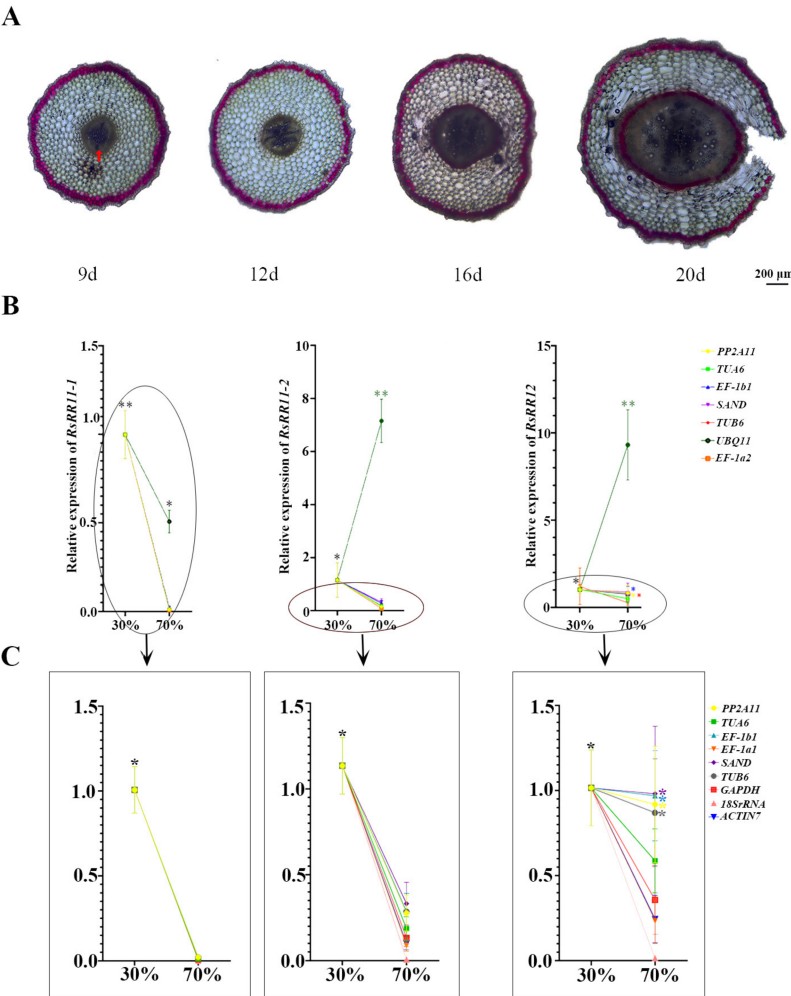

**Figure 4.** Validation of recommended RGs by normalization of *RsRRs* expression in the stele. (**A**) Cross-section diagram of the fleshy root in four periods. The area the red arrow points to is the stele; (**B**) relative expression of *RsRR11-1, RsRR11-2, RsRR12* derived by normalization of seven commonly used RGs; (**C**) relative expression of *RsRR11-1, RsRR11-2, RsRR12* derived by normalization of nine commonly used RGs (*GAPDH, 18SrRNA*, and *ACTIN7* added and *UBQ11* removed) based on Figure 4B. *, $p < 0.05$; **, $p < 0.01$.

**Table 5.** RT-qPCR primer of B-type *RsRRs* genes.

| Gene Name | Gene ID | Primer Sequences (5′-3′) | |
| --- | --- | --- | --- |
| *RsRR11-1* | RSG20107 | F: GGCTTTACCTGAGCAGATTGG | R:GGCTTTGGTGGCGTGAA |
| *RsRR11-2* | RSG25754 | F:GAGCGTAAGGACGGGTTTG | R:TGAGGGAGGTCCAGTTCG |
| *RsRR12* | RSG31240 | F: AGGGTTACCGATGCCTTTAGA | R:TTGAGCAGGAGGAGGAAGAGT |

The results also proved that *PP2A11* is feasible as the RG. Thus, it can be seen that different internal references yield diametrically opposite results.

## 4. Discussion

Cherry radish is a typical root vegetable. The development of cherry radish roots is closely related to its yield and quality. Currently, the mechanism regulating the development and expansion of the fleshy root is not clear. The resolution of the radish expansion mechanism relies on the accurate analysis of gene expression, while the quantitative analysis of gene expression requires the selection of suitable RGs. In cherry radishes, the stele, as the key part for fleshy root expansion, is important for the study of radish expansion. The expanding steles of cherry radishes at different periods were used as materials to screen the RG during fleshy root expansion genes in cherry radishes.

Many studies have shown that the expression of RGs in different tissues, cells, and under different conditions is not always constant, or even varies greatly [35]. So far, no RGs that are consistently expressed under all conditions have been found. There is evidence that the transcript levels of traditional RGs such as *GAPDH* and *ACTIN* may also vary greatly under different experimental conditions or tissue types [36,37]. Therefore, it is important to screen the RG for specific experiments. Currently, studies have shown that the method of screening RGs based on transcriptome data proved to be useful. Narsai et al. used rice tissue, development, and biotic and abiotic transcriptome datasets to identify RGs [38]. Liang et al. screened and evaluated RGs suitable for different tissue samples and/or experimental conditions based on the transcriptome data of *Euscaphis konishii* Hayata [39]. Huang et al., based on transcriptome sequencing, screened *SGT2* and *STK* under cadmium stress of Agaricus antler and showed stronger stability than commonly used RGs [40]. In the study, the FPKM value and gene annotation of transcriptome data were used to initially screen the candidate RGs, and finally, three traditional RGs *GAPDH*, *18SrRNA*, *ACTIN7*, and seven commonly used RGs *UBQ11*, *TUA6*, *TUB6*, *EF-1b1*, *EF-1a2*, *PP2A11*, and *SAND*, were screened out. The stability ranking of the candidate RGs was analyzed and the results showed that the stability of *GAPDH* and *ACTIN7* was not suitable as they did not have the characteristics that were identified as RGs in cherry radish. Some studies have screened the RGs that are used to analyze the expression of radish bolting and flowering-related genes. Xu et al. found, in radish, that *EF-1b1* and *18SrRNA* genes showed great differences in expression under different experimental conditions and were not suitable for RGs, while *TEF2*, *RPII*, and *ACT* were relatively stable and most suitable for RGs [14]. Duan et al. showed that *GAPDH*, *DSS1*, and *UP2* were optimal reference genes for gene expression analysis in all organs and conditions in radish. UPR, GSNOR1, and *ACTIN2/7* were the most stable reference genes in different radish organs [15]. It can be seen that the stability of the same RG is different in the samples under different conditions. Traditional RGs do not have universal adaptation as RGs in different experiments. Therefore, it is necessary to screen RG for specific tissues. The stele tissue of the radish is the key tissue for enlarging and the fleshy root mainly develops from the stele. In the two experimental conditions, the expression of *PP2A11* is the most stable in the stele of cherry radish. Interestingly, the expression of *PP2A11* was also the most stable in the experimental conditions of different lateral roots. *PP2A11* can be recommended as an RG during the growth of cherry radish flesh root.

Recent studies have shown that *PP2A* is the most recommended stable RG for transcriptional normalization in numerous plant tissue samples [41]. In mulberry (*Morus alba* L.), *PP2A* is the preferred RG for the normalization of gene expression data under abiotic stress [42]. The combination of *PP2A*, *SAND*, and *ACTIN* can achieve the normalization of target genes in flower development stages of different color lines [43]. *PP2A*, which is composed of scaffold subunit A, regulatory subunit B, and catalytic subunit C, plays an important role in plant cellular processes [44,45]. Wang et al. also screened out the quantitatively normalized RGs under different stress conditions in *Brassica napus*, among which *TIP41* and *PP2A* were more stable in expression [46]. Tang et al. analyzed the expression stability of RGs of poplar and found that *PP2A-2* and *UBP 1B* (oligouridylate-binding protein 1B) were the best RGs in the regeneration stage of AR (adventitious root) [47]. Notably, *PP2A11* was the most stable gene in both the stele and lateral root (LR) of cherry radish in the current study. These results indicated that the *PP2A* gene may be the specific reference gene in the root development of cherry radishes.

## 5. Conclusions

Taking the gradually expanding stele as material, we screened *PP2A11* as the most suitable RG for the subsequent study of cherry radish root expansion. In contrast, the conventional internal reference genes were not suitable as internal reference genes under this experimental condition. The screening of the suitable RG, *PP2A11*, provides a basis

for analyzing the gene expression during fleshy root development in cherry radishes and facilitates the identification of genes associated with fleshy root enlargement.

**Author Contributions:** R.G. and X.C. conceived the study and revised the manuscript. Y.Y. analyzed the data and wrote the manuscript. Y.Y. and X.W. prepared materials and conducted the experiments. B.C. and S.Z. analyzed data. G.W.-P. helped to revise the manuscript. All authors have read and agreed to the published version of the manuscript.

**Funding:** This research was funded by the Fund for Cooperation Project of Fujian Provincial Science and Technology Program (No. 2022I0010), Youth Academic Training Fund from the College of Horticulture in Fujian Agriculture and Forestry University School of Horticulture (102/722022011), and the Fund for Innovation and Entrepreneurship Projects for College Students (No. 202210389207, No. 202210389035).

**Institutional Review Board Statement:** Not applicable.

**Informed Consent Statement:** Not applicable.

**Data Availability Statement:** The data presented in this study are available on request from the corresponding author.

**Conflicts of Interest:** The authors declare no conflict of interest.

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
