# Peer review of "Evaluation of Reference Genes Suitable for Gene Expression during Root Enlargement in Cherry Radish Based on Transcriptomic Data"

_horticulturae, doi:10.3390/horticulturae9010020_

Round 1

Reviewer 1 Report

The presented article is devoted to the evaluation of suitability of reference genes for analyses of gens’ expression in radish. As a result it was found that PP2A11, is the most suitable reference gene for evaluation of some processes occurring during radish root growth.

At the same time, I have to make some remarks.

First of all, the article does not have chapter “Conclusion”.

Page 2, line 63: There is no explanation of what is HGs.

Page 9, lines 241 – 249 are empty.

Some English language editing is needed.

The article can be published after some corrections and adding of “Conclusion” chapter.

Author Response

Point 1:First of all, the article does not have chapter “Conclusion”

Response 1: We have added the “Conclusion” part (Lines 400-406) and marked them in red in the text and listed them below.

“Taking the gradually expanding stele as material, we screened PP2A11 as the most suitable RG for the subsequent study of cherry radish root expansion. In contrast, the conventional internal reference genes were not suitable as internal reference genes under this experimental condition. The screening of the suitable RG, PP2A11, provides a basis for analyzing the gene expression during fleshy root development in cherry radishes and facilitates the identification of genes associated with fleshy root enlargement.”

Point 2:Page 2, line 63: There is no explanation of what is HGs.

Response 2:HG means housekeeping genes and we have added this information in the text (Line 53-57).

Point 3:Page 9, lines 241 – 249 are empty.

Response 3:We have reformatted the article to conform to the magazine's requirements and the blank lines have been removed. (Line 269-271)

Point 4:Some English language editing is needed.

Response 4:Sorry for the trouble with the language issue, we have reorganized the whole manuscript and asked a native speaker to help with the grammar. Hope that the revision will meet the requirements of the journal and increase the readability of the manuscript.

Reviewer 2 Report

The manuscript attempts to identify better reference genes for gene expression study in the roots of cherry radish (Raphanus sativus), which are essential for accurate quantification of gene expression level. There is a bit of novelty in the manuscript, as it offers several new candidate reference genes and it limits its scope in specific tissues of cherry radish. However, radish is a widely studied plant and similar studies regarding reference genes for gene expression study in radish had been performed, with more sample types and growth conditions (eg. Xu et al. 2012, Duan et al. 2017). Therefore the manuscript has a lot to prove as it should demonstrate that a different and significantly better outcome will be obtained if their candidate reference genes are used in gene expression studies instead of the genes proposed by the other groups. In my opinion the manuscript has largely failed to deliver this, as their validation step indicated mostly similar performance to previous reference genes (figure 4). Thus, the results obtained from this study will have little impact on gene expression studies in cherry radish.

The manuscript also have some major deficiencies in some parts, such as:

1. The tissues used for this study were not described clearly in materials and methods section. Which part of the root was taken? At what stage/time the samples were taken? Are the replicates from different plants or the same plant?

2. Transcriptomic analyses were not adequately described, especially the ones utilizing FPKM values. Is the transcriptome data generated in house? If so, then the steps to obtain the data must be described. If the data was obtained from a published database, then the source (url of transcriptome data and originating paper) must be properly cited and credited, and the exact steps performed to identify the candidate genes must be reported in detail.

3. The authors seems to assume that readers are familiar with utilities like RefFinder, geNorm, etc. Horticulture journal has a wide scope and diverse readers, so niche analyses like those must be accompanied by clear explanations on what the program actually do (its algorithm if necessary), what sort of data were inputed, and how the results should be interpreted. This will also help establish the credibility of the authors as expert in this field.

4. Since several studies have been conducted on this topic, the result should show strong evidences of significant advantages of the newly proposed reference genes to existing reference genes (as indicated by their ranking). In addition, the touted superiority as presented in the discussion section appear to be weakly supported by the data. The highest ranked gene is not much more stable than 18SrRNA, nor more highly expressed (Ct value not too different). As mentioned in the discussion, no RGs that are consistently expressed under all conditions have been found, so differential expression studies typically only focus on genes with higher than double or less than half expression level than control, to reduce the effect of unstable reference and errors. So even if the reference gene is marginally better, it is questionable whether this will impact the way expression studies are conducted.

In terms of writing style, there are also some concerns:

1. English words and grammar styles in some parts make the paper difficult to understand

2. Materials and methods section needs drastic improvement as many experiments are not explained in sufficient detail to allow other research to critically evaluate the result or reproduce the experiments. The worst offenders are the transcriptomic analysis and expression stability analysis.

3. Line 160-162 mentions about 28S and 5S bands, but no supporting figures are presented to back up the claim.

4. Line 175-176 claims good specificity, but figure 1A shows more than one band (although one is clearly fainter than the main band)

5. When categorizing something (eg. line 212 use 100 as the threshold of high expression, or line 215-216 that state that expression is not high) include the citation used to set that criterion, or if the category is self-defined then explain why the threshold values deserve to be set as a threshold value to categorize things.

6. There are two figure 1. For figure 2 there should be explanation on what the numbers on the y axis means. Same explanation needed for figure 3.

7. Is it possible to perform statistical test to see whether differences in relative expression in figure 4 are significant or not? Only UBQ11 show wildly different values. Also, why is the validation in figure 4 limited to just 7 genes instead of 10? Evaluating all 10 genes will make it more in line with other results presented in the paper

8. Line 341-343 need to refer to a table, figure, or citation

9. Line 373-374 seems like a red flag. Genes involved in regulatory function tend to have expression fluctuation, making it less ideal as a reference.

Author Response

Point 1: The manuscript attempts to identify better reference genes for gene expression study in the roots of cherry radish (Raphanus sativus), which are essential for accurate quantification of gene expression level. There is a bit of novelty in the manuscript, as it offers several new candidate reference genes and it limits its scope in specific tissues of cherry radish. However, radish is a widely studied plant and similar studies regarding reference genes for gene expression study in radish had been performed, with more sample types and growth conditions (eg. Xu et al. 2012, Duan et al. 2017). Therefore the manuscript has a lot to prove as it should demonstrate that a different and significantly better outcome will be obtained if their candidate reference genes are used in gene expression studies instead of the genes proposed by the other groups. In my opinion the manuscript has largely failed to deliver this, as their validation step indicated mostly similar performance to previous reference genes (figure 4). Thus, the results obtained from this study will have little impact on gene expression studies in cherry radish.

Response 1: We understand your concern and know that there are already research about the reference gene in radish. However, for the study of root enlargement, no RGs are selected for studying the enlargement of radish roots. Using radish-advanced inbred lines, Xu et al. analyzed the relatively stable expression of genes in different tissue parts, especially in leaves under different conditions, and concluded that RPII (RNA polymerase-II transcription factor), TEF2 (Translation elongation factor 2), and ACTIN are suitable as internal reference genes for quantitative analysis of radish genes [14]. Duan et al. collected the flower buds and siliques of radish at different reproductive stages and obtained UP2 (Uncharacterized conserved protein UCP022280) and GAPDH (Glyceraldehyde-3-phosphate dehydrogenase) as suitable RG for radish pistil development studies [15]. Selection of RG was also conducted in different organs including root, stem, leaf, as well as calyx, petal, stamen, and pistil and UPR (Uncharacterized protein family), GSNOR1 (GroES-like zinc-binding dehydrogenase family protein), and ACTIN2/7 were the most stable internal reference genes in radish [15].

We can clearly see the reported RG is selected mainly from leaves or reproductive organs. It is quite different from the root. The fleshy root of the radish mainly develops from the stele [16]. In the current study, the enlarging part of the cherry radish (stele) was used to screen the RGs for analysis of the gene expression during the enlargement of the radish, which will give a better choice for RG selection in studying root enlargement.

Point 2: The tissues used for this study were not described clearly in materials and methods section. Which part of the root was taken? At what stage/time the samples were taken? Are the replicates from different plants or the same plant? 

Response 2: Sorry for the problem. We have added more information about the sampling process in the Materials and methods part and listed it below (Lines 107-121). We have also added figures about the radish root cross-section in Fig. 4A and the stele is what we take for the qRT-PCR analysis.

“The material for this experiment was the gradually expanding stele of cherry radish. Twelve radish seedlings were planted in one hydroponic container, 30 containers were planted at a time, and from day 9 onwards, three seedlings were taken from each of the three containers for sectioning, and the rest were used for sampling and analysis. The specific observation and sampling process were as follows: after the radish was transplanted (9 d) until the cherry radish cortex rupture (20 d), the root of the cherry radish was observed daily, and samples were taken, sliced, and microscopically examined to calculate the ratio of the diameter of the radish stele cross-section to the diameter of the root, and thus analyze the expansion of the cherry radish root. Finally, 12-day- and 20-day-old roots with a ratio of 30% and 70%, respectively, were collected for the qRT-PCR analysis. Three biological replicates were performed for each experiment. When sampling the stele, the enlarged part of the radish was broken in the middle and the cortex was split, while the stele remained intact. Then the external cortex was removed and the stele was cut off with a clean blade, weighed and loaded into a lyophilization tube, and quickly placed in liquid nitrogen. Three biological replicates were performed for each experiment.”

Point 3: Transcriptomicanalyses were not adequately described, especially the ones utilizing FPKM values. Is the transcriptome data generated in house? If so, then the steps to obtain the data must be described. If the data was obtained from a published database, then the source (url of transcriptome data and originating paper) must be properly cited and credited, and the exact steps performed to identify the candidate genes must be reported in detail.

Response 3: Thank you for reminding us. The transcriptomic data were obtained from a database with BioProject ID PRJNA874325 (https://dataview.ncbi.nlm.nih.gov/object/PRJNA874325?reviewer=r6vc02uh53bundl6rarngrsds.) (Lines 140-143). Based on the previous studies on cruciferous RGs, we performed gene function annotation on the RGs in transcriptome data and initially screened out 23 different functional candidate RGs. By analyzing the gene FPKM values, and general PCR analysis, we found that the FPKM values of 13 of these RGs were not high, and running PCR with mixed samples revealed that the bands were not bright, and finally 10 RGs were identified as candidates (Lines 143-150).

Point 4: The authors seems to assume that readers are familiar with utilities like RefFinder, geNorm, etc. Horticulture journal has a wide scope and diverse readers, so niche analyses like those must be accompanied by clear explanations on what the program actually do (its algorithm if necessary), what sort of data were inputed, and how the results should be interpreted. This will also help establish the credibility of the authors as expert in this field.

Response 4: Thank you for your suggestion. We have added more information about the software (Lines 174-184). The expression stability of candidate genes was evaluated by BestKeeper, Norm Finder, and geNorm from RefFinder [13,19]. The analysis principle of geNorm is calculating the stability value M by entering the Ct value [20]. The criterion is that the smaller the M value, the better the stability of reference genes; otherwise, the worse the stability of reference genes. The software NormFinder was used to analyze the stability of internal reference genes [21]. The stable value was calculated and the variation between samples was analyzed. BestKeeper software was used to determine the stability ranking of reference genes by analyzing the correlation coefficient, standard deviation, and coefficient of variation among each gene [22]. The greater the correlation coefficient, the smaller the standard deviation and the coefficient of variation, showing the better the stability of the reference gene, and vice versa.

Point 5: Since several studies have been conducted on this topic, the result should show strong evidences of significant advantages of the newly proposed reference genes to existing reference genes (as indicated by their ranking). In addition, the touted superiority as presented in the discussion section appear to be weakly supported by the data. The highest ranked gene is not much more stable than 18SrRNA, nor more highly expressed (Ct value not too different). As mentioned in the discussion, no RGs that are consistently expressed under all conditions have been found, so differential expression studies typically only focus on genes with higher than double or less than half expression level than control, to reduce the effect of unstable reference and errors. So even if the reference gene is marginally better, it is questionable whether this will impact the way expression studies are conducted.

Response 5:18SrRNA, GADPH, and ACTIN have been used as candidate reference genes in a variety of experiments. The three genes have been validated for stability under a variety of material experimental conditions, and for this reason, they are often used as traditional internal reference genes for studies comparing and analyzing the stability of other common housekeeping genes.

The previous analysis of the expression of 18SrRNA showed that the expression of the traditional reference gene was not high in the stele. To compare the stability difference between the traditional reference gene and the common housekeeping genes, the three reference genes of 18SrRNA, GADPH and ACTIN were continued as candidate genes for comparison. It was found that 18SrRNA was relatively stable, but it was still difficult to be used as a reference gene due to its low expression. It is concluded that even the traditional internal reference genes are not equally applicable under experimental conditions. However, PP2A11 not only has a higher expression level than 18SrRNA but also has higher stability than 18SrRNA. 

Point 6: English words and grammar styles in some parts make the paper difficult to understand.

Response 6: Sorry for the problems caused by the language. We have reorganized the whole manuscript and asked a native speaker to help with the grammar. Hope that the revision will meet the requirements of the journal and increase the readability of the manuscript.

Point 7: Materials and methods section needs drastic improvement as many experiments are not explained in sufficient detail to allow other research to critically evaluate the result or reproduce the experiments. The worst offenders are the transcriptomic analysis and expression stability analysis.

Response 7: We have revised the material and methods parts and added more detailed information about the sampling and analysis.

Point 8 : Line160-162 mentions about 28S and 5S bands, but no supporting figures are presented to back up the claim.

Response 8: We have updated the figure and revised the description for better understanding (Lines 189-191).

Point 9: Line175-176 claims good specificity, but figure 1A shows more than one band (although one is clearly fainter than the main band)

Response 9: Figure 1A is an electrophoretic running gel plot of RNA, with the typical two strong bands. It is not a PCR product of a gene.

Point 10: When categorizing something (eg. line 212 use 100 as the threshold of high expression, or line 215-216 that state that expression is not high) include the citation used to set that criterion, or if the category is self-defined then explain why the threshold values deserve to be set as a threshold value to categorize things.

Response 10: It is a misunderstanding. The statement is intended to demonstrate relative levels of expression. We have modified the statements (Lines 228-231). 

Point 11: There are two figure 1. For figure 2 there should be explanation on what the numbers on the y axis means. Same explanation needed for figure 3.

Response 11: We have revised and added explanations for Figures 2 and 3.

Point 12: Is it possible to perform statistical test to see whether differences in relative expression in figure 4 are significant or not? Only UBQ11 show wildly different values. Also, why is the validation in figure 4 limited to just 7 genes instead of 10? Evaluating all 10 genes will make it more in line with other results presented in the paper.
Response 12: Thank you for your opinion. We have analyzed the significant difference by using the single factor ANOVA of the mean value analysis method in SPSS software. It showed UBQ11 had higher expression variation. The results indicated that there was still variable expression even for relatively stable HGs [32]. Moreover, the seven genes were selected for verification because of the conclusion drawn above: the expression levels of the other three traditional genes were low. The stability analysis of the three traditional genes (18SrRNA, GAPDH, and ACTIN) was made to compare the differences between the three traditional genes and the ordinary reference genes in this experiment. The results indeed concluded that the traditional reference genes may not always be the reference genes (compared with research by Xu et al. 2012, and Duan et al. 2017), especially on the stele in cherry radish. The other two genes (GAPDH and ACTIN) were also very unstable and were not tested with these three genes (Lines 231-236, 281-285) .

Point 13:  Line341-343 need to refer to a table, figure, or citation

Response 13: We have referred citations (Narsai et al., 2010) (Liang et al., 2018). (Lines 256-258) . 

Point 14: Line373-374 seems like a red flag. Genes involved in regulatory function tend to have expression fluctuation, making it less ideal as a reference.

Response 14: We have modified the statement (Lines 293-299)

Reviewer 3 Report

Dear Authors,

I find your research very interesting; you have not mentioned how your research can be useful and its application in the article. If you can provide those details, it will be helpful for the readers as well as research who will follow this article.

Regards

Author Response

Point1: I find your research very interesting; you have not mentioned how your research can be useful and its application in the article. If you can provide those details, it will be helpful for the readers as well as research who will follow this article.

Response1: Thank you for your affirmation and valuable comments on our research.

The fleshy root is the storage and edible organ of cherry radish, and the development of the fleshy root is closely related to the yield and quality of cherry radish. Taking the gradually expanding stele as material, we screened PP2A11 as the most suitable RG for the subsequent study of cherry radish root expansion. The screening of the suitable RG, PP2A11, provides a basis for analyzing the gene expression during fleshy root development in cherry radishes. This research provides a useful and reliable RGs resource for the accurate study of gene expression during root enlargement in cherry radishes and facilitates the functional genomics research on root enlargement.

We have also added this information in the abstract (Lines 28-30) and text (Lines 36-39; 389-394).

Reviewer 4 Report

The manuscript is really important as the reference genes may change according to different conditions that we work with plants. Then, all the sequences that the authors check is really important. And it was a question raised in a article in plant journal which would be the good reference gene to use in qRT-PCR.

The manuscript idea is really important, but it need to be improved and it need to have english checked by a native speaker.

1) Abstract need to follow journal model, without headling. It need to improve. It is difficulto to understand.

2) keywords - usually are different from title, please modify.

3) revise english by a native speaker it is really difficult to understand. Be careful with paragraph built - for example line 38 The formation ...

line 39  The formation and....

line 39  process of metamorphosis. The formation and development of metamorphic

Be careful, it makes to difficult to read and understand the ideias 

4)  references in the text are not by numbers, they did not follow journal instructions

5) methods  - it is not clear the controls that were make to be sure that there are no DNA contamination on RNA - it is important to follow the article " Eleven golden rules of quantitative RT-PCR from Udvard et al. 2008".

6) Results - I did not understand figure 1 and the text line 160 to line 162 " Electrophoresis detection showed that the 28S and 18S bands were clear, and the brightness of the 28S band was about twice that of the 18S band, while the brightness of the 5S band was lower" - I did not see on gel these bands, fig 1a showed the PCR results. Then it was clear for me what the authors means about the 28S and 18S as the gel is about PCR.

7) in general the ideias need to be developed in writing, it gives the impression that many ideas were put together without conecting or developing.

Author Response

The manuscript is really important as the reference genes may change according to different conditions that we work with plants. Then, all the sequences that the authors check is really important. And it was a question raised in a article in plant journal which would be the good reference gene to use in qRT-PCR.

The manuscript idea is really important, but it need to be improved and it need to have english checked by a native speaker.

Point 1: Abstract need to follow journal model, without headling. It need to improve. It is difficult to understand.

Response 1: Thank you for your advice. We have changed the abstract following the journal’s requirements (Lines 17-31) and listed it below.

“Reliable reference genes (RGs) are of great significance for the normalization of quantitative data. RGs are often used as a reference to ensure the accuracy of experimental results to detect gene expression levels by reverse transcription-quantitative real-time PCR (RT-qPCR). To evaluate the normalized RGs which are suitable for studying the expression of genes during the process of radish stele enlargement. Based on the functional annotations and fragment per kilobase of transcript per million mapped reads (FPKM) values in the transcriptome data, three traditional RGs (GAPDH18SrRNA, and ACTIN7) and seven commonly used RGs (UBQ11TUA6TUB6EF-1b1EF-1a2PP2A11, and SAND) were obtained. In the study, the results of geNorm, NormFinder, and Bestkeeper from RefFinder comprehensively analyzed the stability ranking of candidate RGs. The results showed that compared with the traditional RGs, the common RGs show higher and more stable expression. Among the seven commonly used RGs, PP2A11 is recommended as the optimal RG for studying cherry radish stele enlargement. This research provides a useful and reliable RGs resource for the accurate study of gene expression during root enlargement in cherry radishes and facilitates the functional genomics research on root enlargement.”

Point 2: keywords - usually are different from title, please modify.

Response 2: We have revised the keyword to “Cherry radish; root enlargement; stele; reference genes; PP2A11.” (Lines 32)

Point 3: revise english by a native speaker it is really difficult to understand. Be careful with paragraph built - for example line 38 The formation ...

line 39 The formation and....

line 39 process of metamorphosis. The formation and development of metamorphic

Be careful, it makes to difficult to read and understand the ideias

Response 3: Sorry for the problems caused by the language. We have reorganized the whole manuscript and asked a native speaker to help with the grammar. Hope that the revision will meet the requirements of the journal and increase the readability of the manuscript.

Point 4: references in the text are not by numbers, they did not follow journal instructions

Response 4: We have modified the format of the references according to the requirements of the journal.

Point 5: methods - it is not clear the controls that were make to be sure that there are no DNA contamination on RNA - it is important to follow the article " Eleven golden rules of quantitative RT-PCR from Udvard et al. 2008".:

Response 5:Thank you for the suggestion. We have followed and cited the suggested reference. DNA is removed. We digested purified RNA with gDNA Clean Reagen to remove contaminating genomic DNA (Lines 128). The experiment was carried out in a clean and dust-free fume hood. The experimental instruments were treated with high-pressure asepsis. Researchers were required to wear rubber gloves and mask during the experiment to avoid contamination of RNA (Lines 135-137).

Point 6:Results - I did not understand figure 1 and the text line 160 to line 162 " Electrophoresis detection showed that the 28S and 18S bands were clear, and the brightness of the 28S band was about twice that of the 18S band, while the brightness of the 5S band was lower" - I did not see on gel these bands, fig 1a showed the PCR results. Then it was clear for me what the authors means about the 28S and 18S as the gel is about PCR.c:

Response 6:In this study, we just used agarose gel electrophoresis combined with measuring the OD280/OD260 ratio to detect the quality of the extracted RNA. To ensure accuracy, we performed electrophoresis again using the RNA, and the 28S and 18S bands were still clearly visible, we also replaced the new electrophoresis graph in Fig. 1 and labeled the bands, and explained the meaning of the bands in the figure notes.

Point 7:in general the ideias need to be developed in writing, it gives the impression that many ideas were put together without conecting or developing.

Response 7:We have reorganized the whole manuscript and revised the introduction and discussion part to better convey the information. Based on the transcriptomic data of cherry radish, ten candidate RGs were selected in the current study. We analyzed the expression stability of candidate RGs and evaluated the specificity of RGs by identification of reliable RGs and multilevel analysis. Finally, we validated the RGs expression normalization with b-type ARR.

Round 2

Reviewer 2 Report

The manuscript is significantly improved compared to the previous version. However, there are still some major issues that have not been addressed properly:

1. Claim of low expression of traditional reference genes like 18SrRNA (in abstract, line 223-226, and other parts of the manuscript), is not supported by data presented in the paper. Where did this conclusion come from? All data, presented in Figure 1B, 1C, and table 3, indicate comparable brightness and Ct values between the traditional references and the selected RG (PP2A11). Unless the low expression claim is removed then the selected RG is clearly unsuitable as well because it has a low expression level too. Is it possible that the low expression claim came from FPKM value of initial transcriptome data? If so, the FPKM data should be included in Table 1 as a column. But even if this was true, it is not logical to stick to results obtained by others while every results obtained by the authors indicated otherwise.

2. Figure 4B is crucial for this paper, as it provides the results of the first real test to see if the selected RG is truly superior in the analysis of genes of interest. So ideally it highlights the differences between this RG and others, including traditional RGs as those are the ones most often used by others. It is clear that the current figure 4B has two problems: missing data from the 3 traditional RGs, and unusually low expression of UBQ11 at 70% masking the differences between PP2A11 and others. To address this, I suggest that Figure 4C is added, containing the same data as 4B with UBQ11 removed, to highlight the contrast between PP2A11 and the rest, along with statistical test to indicate if there are significant differences between PP2A11 and other RGs.

3. For the stability analysis, 2 groups of samples were evaluated: stele and lateral roots. For the stele group, the samples came from 12 day roots, 20 day roots, or both? If both sample types were used, why was UBQ11 expression so inconsistent when used as RG to test the RR genes?

4. Ranking from BestKeeper (table 3) is unintuitive. Logically, genes with the smallest SD and lowest CV should be the most stable, but it seems that the software places more importance on other parameters or calculation that is not present in the table. That parameter should be included in the table so that the ranking makes more sense. Alternatively, the ranking system should be explained better either in the methods section or results section.

5. Explanation of the way stability ranking software works in the methods section is insufficient. For example, the explanation of the mechanism of genorm as "The criterion is that the smaller the M value, the better the stability of reference genes; otherwise, the worse the stability of reference genes" does not explain how the software works. A more proper way to explain it, quoting directly from the original genorm paper, is "the internal control gene-stability measure M is the average pairwise variation of a particular gene with all other control genes. Genes with the lowest M values have the most stable expression. Assuming that the control genes are not co-regulated, stepwise exclusion of the gene with the highest M value results in a combination of two constitutively expressed housekeeping genes that have the most stable expression in the tested samples." But of course the original paper must not be quoted verbatim. That quoted sentence("The criterion is that....") fits better in the results section, when explaining the figure or table from the software outputs.

6. The manuscript title probably should use 'Evaluation of' or 'Identification of' instead of Evaluate.

Author Response

The manuscript is significantly improved compared to the previous version. However, there are still some major issues that have not been addressed properly:

Point 1: Claim of low expression of traditional reference genes like 18SrRNA (in abstract, line 223-226, and other parts of the manuscript), is not supported by data presented in the paper. Where did this conclusion come from? All data, presented in Figure 1B, 1C, and table 3, indicate comparable brightness and Ct values between the traditional references and the selected RG (PP2A11). Unless the low expression claim is removed then the selected RG is clearly unsuitable as well because it has a low expression level too. Is it possible that the low expression claim came from FPKM value of initial transcriptome data? If so, the FPKM data should be included in Table 1 as a column. But even if this was true, it is not logical to stick to results obtained by others while every results obtained by the authors indicated otherwise.

Response 1: We understand your concern. As you said, the low expression of 18SrRNA, ACTIN7, and GAPDH is a result of the transcriptome data (root steles of 12, 16, and 20 days) and their FPKM values are 21.8; 24.4; and 5.4, respectively. These three genes are at a low expression compared to other candidate internal reference genes with FPKM values above 90. We have also added the corresponding FPKM values in Table 1. This was also found when the cherry radish steles at different developmental stages (9, 12, 16, and 20 days) as mixed samples for analysis. In Figure 1B, We used Image J software to calculate the gray value of the bands. Results showed the mean gray value of 10 genes in Figure B is EF-1a2 (15786.619), EF-1b1 (14097.669), SAND (13025.205), UBQ11 (12611.255), ACTIN7 (11754.255), PP2A11 (9059.305), TUA6 (7496.012), 18SrRNA (6984.305), TUB6 (6154.891), GAPDH (4413.77), the brightness of the bands corresponding to 18SrRNA, ACTIN7, and GAPDH genes indicates their transcripts were at low levels among the tested ten genes (Lines 198-199). The CT values of the 18SrRNA, ACTIN7, and GAPDH were 30.04, 29.92, 29.69, respectively, in the average expression levels of the samples shown in Figure 1C, indicating that their expression level was not high in the stele among the tested ten genes. Taken together, these results suggest that 18SrRNA, ACTIN7, and GAPDH are not the best internal references for studying gene expression during radish expansion.

To make the presentation clearer, we have modified the description of "low expression" as suggested. As for the difference between this result and those of other studies, it may be due to the specificity of the species radish. Radish undergoes metamorphic development in the stele of the root and expand to a big size. There are no studies that have been conducted to screen for internal reference genes in radish root expansion studies. This special botanical characteristic of radish also makes its expanding part as its edible organ. It is possible that during the development of this expanding organ, there are different gene expression characteristics than in other root, stem, leaf, flower, fruit, and seeds, which was also confirmed in our study, so these traditional genes (18SrRNA, ACTIN7, and GAPDH) are not suitable as internal reference for studying radish expansion and need to be screened separately. This is also the reason why there are no internal reference genes that are widespread and applicable to all developmental processes. 

Point 2 :  Figure 4B is crucial for this paper, as it provides the results of the first real test to see if the selected RG is truly superior in the analysis of genes of interest. So ideally it highlights the differences between this RG and others, including traditional RGs as those are the ones most often used by others. It is clear that the current figure 4B has two problems: missing data from the 3 traditional RGs, and unusually low expression of UBQ11 at 70% masking the differences between PP2A11 and others. To address this, I suggest that Figure 4C is added, containing the same data as 4B with UBQ11 removed, to highlight the contrast between PP2A11 and the rest, along with statistical test to indicate if there are significant differences between PP2A11 and other RGs.

Response 2: This is a very good suggestion to make the results clearer. As suggested, we have removed the expression of UBQ11 and added the expression of three internal reference genes, and also performed the corresponding significance analysis (Line 344-353). The results showed that the differences between RsRR11-1 and RsRR11-1 at different periods of the stele development could be shown when nine candidate genes were used as an internal reference, while for RsRR12, the expression was significant when three traditional genes (18SrRNA, ACTIN7, and GAPDH) were used as the internal reference. While expression was insignificant when PP2A11 was calculated as the internal reference. Thus, it can be seen that different internal references yield diametrically opposite results. 

Point 3: For the stability analysis, 2 groups of samples were evaluated: stele and lateral roots. For the stele group, the samples came from 12 day roots, 20 day roots, or both? If both sample types were used, why was UBQ11 expression so inconsistent when used as RG to test the RR genes?

Response 3: Regarding the stability analysis of the internal reference genes, two groups of samples were selected for the experiment, one for the stele part, i.e. group I, and one for the lateral roots, group 2, both at 9 d, 12 d, 16 d and 22 d for cherry radish. The figures on the left side of the ABC are shown in Figure 2 to demonstrate the stability of each candidate endogenous gene in group I, and on the right side is the stability of each candidate internal reference gene in group II. Since the data of group I and group II are from different samples, it is normal that the expression of UBQ11 appears inconsistent. This is because the stele underwent a metamorphosis, which is the main site of radish expansion, a phenomenon not found in the model plant, while the lateral root is a normal organ and no metamorphosis occurred. The results of stability analysis showed that the expression of the internal reference genes was different in different tissue sites, especially in UBQ11, where stability was good in the lateral roots but not in the stele, this might be because the gene expression of UBQ11 was related to the metamorphic development of the stele.

To make the results easier to understand, we have explained in the Materials Methods section, what kind of analysis the samples were sampled for in different periods and marked them in red (Lines 117-119).

Point 4: Ranking from BestKeeper (table 3) is unintuitive. Logically, genes with the smallest SD and lowest CV should be the most stable, but it seems that the software places more importance on other parameters or calculation that is not present in the table. That parameter should be included in the table so that the ranking makes more sense. Alternatively, the ranking system should be explained better either in the methods section or results section.

Response 4: This section has been re-edited to show the analysis method more clearly and marked in red (Line 283). The stability values of the genes were also added in Table 3. Specifically: According to the method written and analyzed by Michael et al. (2004) [19], this program calculates the correlation coefficient (r), standard deviation (SD), and coefficient of variation (CV) of the pairing generated between each gene by entering the Ct value of the gene. The determination is then made by comparing the magnitude of each value. The principle is that the larger the correlation coefficient, the smaller the standard deviation and the coefficient of variation, the better the gene stability (Table 3) value of the internal reference gene, and vice versa. 

Point 5: Explanation of the way stability ranking software works in the methods section is insufficient. For example, the explanation of the mechanism of genorm as "The criterion is that the smaller the M value, the better the stability of reference genes; otherwise, the worse the stability of reference genes" does not explain how the software works. A more proper way to explain it, quoting directly from the original genorm paper, is "the internal control gene-stability measure M is the average pairwise variation of a particular gene with all other control genes. Genes with the lowest M values have the most stable expression. Assuming that the control genes are not co-regulated, stepwise exclusion of the gene with the highest M value results in a combination of two constitutively expressed housekeeping genes that have the most stable expression in the tested samples." But of course the original paper must not be quoted verbatim. That quoted sentence("The criterion is that....") fits better in the results section, when explaining the figure or table from the software outputs.

Response 5: To better illustrate the analysis process of stability, we have cited the original document and reorganized the principles of determination for the three software as suggested, and also updated the information in the text and marked it in red (Lines 251-274). The list is as follows.

geNorm is a software that calculates the stability M value of each RG by entering the Ct value in the program to filter out the RGs with better stability [26]. geNorm (M) provides a comparison of stability as the average pairwise variation between the gene and all other candidate genes [23]. Therefore, the principle of its judgment is that if the gene exhibits a high degree of variability among samples i.e., the higher the M value, the less stable the gene is, the less suitable it is for RG selection, and vice versa, the better it is for RG selection. Norm Finder focuses on finding genes that have less variation in expression within and between groups, and then the gene with the least variation is measured. The criteria for the NormFinder program are similar to those for the geNorm program. The principle is that the smaller the stability M-value, the more stable the gene expression is, and the worse the opposite [27]. BestKeeper is a method based on stability index (BKI) calculations that indicates the highest stability [19]. This software also performs the analysis by directly entering the Ct values of the samples. By calculation, BestKeeper can obtain the correlation coefficient (r), standard deviation (SD), and coefficient of variation (CV) between each gene to generate pairs and then compare the values to calculate the stability value [29]. The criterion principle is that the larger the correlation coefficient (r), the smaller the standard deviation (SD), and the coefficient of variation (CV), the better the stability value of the internal reference genes; conversely, the worse the stability, and the final determination of the RG stability ranking [19].

Point 6: The manuscript title probably should use 'Evaluation of' or 'Identification of' instead of Evaluate. 

Response 6: Thanks and we have revised the title as suggested (Line 2).

Reviewer 4 Report

The authors made the modifications suggested. Worked in the text, add information and figures. The manuscript improved a lot.

As I mention before, the scientific data is really important for qRT-PCR

The authors need to check some paragraphs as below:

Line 343 the study of radish expansion. In this study, we used the expanding steles of cherry

line 344 radishes at different periods as materials to screen the RG to study the expression of ....

Author Response

The authors made the modifications suggested. Worked in the text, add information and figures. The manuscript improved a lot.

As I mention before, the scientific data is really important for qRT-PCR

Point 1: The authors need to check some paragraphs as below:

Line 343 the study of radish expansion. In this study, we used the expanding steles of cherry

line 344 radishes at different periods as materials to screen the RG to study the expression of ....

Response 1: Thanks for your critical reading. We have reorganized these paragraphs and marked them in red (Lines 370-371).

Round 3

Reviewer 2 Report

The manuscript has improved considerably and with minor revisions it will be ready for publication. Some minor adjustments that may improve the manuscript further:

Line 148: add reference to Table 1.

Line 231: Table 1 title may be more appropriate to use "Selected candidate reference genes and their respective PCR primers" as the addition of FPKM data is a bit out of line with the old title

Line 324: please find the proper scientific words to describe the phenomenon

Line 330: since RNAseq (FPKM) data is mentioned while referring to Figure 4B, it must be included as one of the data points in figure 4B & 4C

Line 338-346: When calculating significance, is it based on one-on-one comparison or general comparison? Please indicate which approach was taken. I also think that it will be informative to use FPKM as the baseline and compare all others to it in figure 4C, since it is a different method than RG in QPCR. It is of course debatable whether FPKM is the most accurate, but it is useful to have this information presented to allow the readers to make their own decision for their future experiment.

Author Response

The manuscript has improved considerably and with minor revisions it will be ready for publication. Some minor adjustments that may improve the manuscript further:

Point 1:Line 148: add reference to Table 1.
Response 1: We have referred to Table 1 in Line 148.

Point 2:Line 231: Table 1 title may be more appropriate to use "Selected candidate reference genes and their respective PCR primers" as the addition of FPKM data is a bit out of line with the old title.

Response 2: We have changed it to “Selected candidate reference genes and their respective PCR primers” (Line 235).

Point 3: Line 324: please find the proper scientific words to describe the phenomenon

Response 3: We have revised the sentence as below:

The process of radish expansion is shown in Figure 4A. With the extension of time, the radish's stele area expands laterally at a faster rate, eventually causing the rupture of the cortex on the 20th day and the typical "belly-breaking" phenomenon (Line 324-327).

Point 4:  Line 330: since RNAseq (FPKM) data is mentioned while referring to Figure 4B, it must be included as one of the data points in figure 4B & 4C.

Response 4: Regards to the FPKM data, we have removed the relevant expressions (Line336-345) as the RNAseq data have been used in another study.

We have cited the original database in this MS  (https://dataview.ncbi.nlm.nih.gov/object/PRJNA874325?reviewer=r6vc02uh53bundl6rarngrsds).

Point 5: Line 338-346: When calculating significance, is it based on one-on-one comparison or generalcomparison? Please indicate which approach was taken. I also think that it will be informative touse FPKM as the baseline and compare all others to it in figure 4C, since it is a different method than RG in QPCR. It is of course debatable whether FPKM is the most accurate, but it is useful to have this information presented to allow the readers to make their own decision for their future experiment.

Response 5: The significant difference is based on a one-to-one comparison using SPSS software. We have added this information to the method part (Line 187-188).

It is a good idea to use FPKM as the baseline and do the comparison, but the data of FPKM value of RsRR genes can not be used again as we have explained in the last question. Based on the current data, it is clear which gene is suitable for the reference gene in the study of radish root expansion.